# GEOMETRIC DIMENSIONALITY REDUCTION: FRAMEWORK AND OPEN CHALLENGES

## ABSTRACT

The efficient processing of increasingly feature-dense data has become a critical area of research in both industry and academia. Applications such as data visualization, embedded neural network inference, and reducing computational complexity are founded upon the ability to project data into lower dimensions with minimal neighborhood distortion. Current statistical algorithms such as UMAP, t-SNE, and LLE utilize assumptions regarding data distribution and non-trivial hyperparameter tuning. Consequently, out-of-sample projections are non-deterministic and the corresponding reduced axis are non-interpretable. In this paper we propose the framework of Geometric Dimensionality Reduction, a novel technique utilizing algebraic and topological symmetries to semi-reversibly & logarithmically compress data. Preconditioning metrics for reducing projection distortion, improving clustering accuracy, and dimension reversibility are proposed as well. Additionally, we demonstrate a closed-form projection for $\mathbb{R}^4 \mapsto \mathbb{R}^3$ and compare results with UMAP and t-SNE. Finally, we discuss open challenges and provide a complete framework for $\mathbb{R}^{2^n} \mapsto \mathbb{R}^{n+1}$ dimensionality reduction.

## 1 INTRODUCTION

Non-linear dimensionality reduction is currently dominated by algorithms such as t-SNE and UMAP. Beyond data visualization, these frameworks offer the ability to compress feature-dense data into samples suitable for training the most efficient machine learning models (McInnes et al., 2018; van der Maaten & Hinton, 2008). While both methods aim to best preserve global and local spatial structure, drawbacks include the inability or increased complexity of projecting out-of-sample data into the low-dimensional embedding, as well as the inability to re-project compressed data into the original high-dimensional space. Furthermore, producing meaningful projections often requires assumptions about data distribution and tuning non-trivial, dataset-specific hyperparameters (Lin & Fukuyama, 2024).

Geometric Dimensionality Reduction offers a novel framework for non-linear data compression that utilizes low dimensional manifolds, with convenient algebras or topologies, embedded in high dimensional spaces. Algorithmic paradigms of this form typically consist of two maps: contraction, and projection. The first is a map from the vectorization of each element in the set to its closest point on the manifold, and the second is a projection from the manifold onto a related surface embedded in a lower dimension (Livschitz & Gu, 2022). Utilizing the continuous structure of the manifold, discrete optimization problems such as minimizing the distance between two sets are replaced with a continuous differential equation for minimizing the distance between points in a set and a surface. Additional features such as; preconditioning, partial compression reversibility, and computational efficiency improvements, make Geometric Dimensionality Reduction a compelling alternative to current state-of-the-art methods.

In this paper we provide a fully worked example for compressing $\mathbb{R}^4 \mapsto \mathbb{R}^3$ using a novel Geometric Dimensionality Reduction framework entitled LGE. We generalize the LGE framework for $\mathbb{R}^{2^n} \mapsto \mathbb{R}^{n+1}$, followed by a review of how contemporary methods compare to LGE, and complete our analysis with a summary of open problems and proposed solutions.

## 2  MATHEMATICAL BASIS

The LGE technique for logarithmic data compression is defined by two distinct maps. The first is a contraction from $S^{2^n-1}$ onto $\mathbb{S}^{2^n-1}_{\mathbb{G}_n}$, and the second is a projection from $\mathbb{S}^{2^n-1}_{\mathbb{G}_n}$ onto $S^n$. The specific mathematical tools used in the construction of this technique are: methods of solving non-linear equations, polyspherical coordinates, and Multicomplex numbers (Luna-Elizarrarás et al., 2015). The generalized framework around formulating and solving for the contraction map $\mathbf{C}$ will be provided in this paper, while the projection map $\mathbf{P}$ is derived in Livschitz & Gu (2022). In addition to the two maps that jointly form LGE dimensionality reduction, a number of metrics that optimize clustering fidelity are also proposed. The underlying composition of maps that define the algorithm are highlighted in the following diagram:

$$\mathbf{LGE}(\vec{x}) = \|\vec{x}\| \cdot \mathbf{P}\big(\mathbf{C}\big(\frac{\vec{x}}{\|\vec{x}\|}\big)\big), \;\; \vec{x} \in \mathbb{R}^{2^n} \tag{1}$$

$$\mathbb{R}^{2^n} \xrightarrow{\frac{\mathbf{x}}{\|\mathbf{x}\|}} S^{2^n-1} \xrightarrow{\;\mathbf{C}\;} \mathbb{S}^{2^n-1}_{\mathbb{G}_n} \xrightarrow{\;\mathbf{P}\;} S^n \xrightarrow{\|\mathbf{x}\| \cdot \frac{\mathbf{x}'}{\|\mathbf{x}'\|}} \mathbb{R}^{n+1} \tag{2}$$

$$\mathbb{S}^{2^n-1}_{\mathbb{G}_n} \subset S^{2^n-1} \subset \mathbb{R}^{2^n}, \; S^n \subset \mathbb{R}^{n+1}, \; \vec{x} \in \mathbb{R}^{2^n}, \; \vec{x'} \in \mathbb{R}^{n+1} \tag{3}$$

We begin the derivation of the contraction map $\mathbf{C}$ by defining the Simple Multicomplex Rotation Group $\mathbb{S}^{2^n-1}_{\mathbb{G}_n}$. This multiplicative group is generated by the product of $n$ commuting independent complex rotations, which results in a manifold constrained onto surface of $S^{2^n-1}$ after vectorization. We firstly must define the algebraic properties of the Simple Multicomplex Units that will be used to generate the rotation group (Luna-Elizarrarás et al., 2015):

**Definition 1 (Simple Multicomplex Units)** *Let* $i_k \neq i_m$ *iff* $k \neq m$, $i_m i_k = i_k i_m$, $1 \leq m, k \leq n$

$$\mathbb{G}_n \equiv \big\{ i_k \,|\, i_k^2 = -1, \; 1 \leq k \leq n \big\} \tag{4}$$

Similar to Euler's formula for defining the complex unit circle, we can now use the Simple Multicomplex Units to construct the Simple Multicomplex Rotation Group (Price, 2018). $\mathbb{S}^{2^n-1}_{\mathbb{G}_n}$ can be defined as the product of $n$ distinct simple Multicomplex rotations, which forms the following $n$-dimensional manifold:

**Definition 2 (Simple Multicomplex Rotation Group)** *Let* $\theta_1 \in [0, 2\pi)$, $\theta_k \in [0, \pi)$, $2 \leq k \leq n$

$$\mathbb{S}^{2^n-1}_{\mathbb{G}_n} \equiv \Big\{ \prod_{k=1}^{n} e^{i_k \theta_k} \,|\, i_k \in \mathbb{G}_n \Big\} \tag{5}$$

$$= \Big\{ \prod_{k=1}^{n} \cos(\theta_k) + \sum_{k=1}^{n} i_k \bigg( \sin(\theta_k) \prod_{\ell=k+1}^{n} \cos(\theta_\ell) \prod_{m=1}^{k-1} e^{i_m \theta_m} \bigg) \,|\, i_k \in \mathbb{G}_n \Big\} \tag{6}$$

**Definition 3 (General LG fibration Projection)** $i_k^2 = -1$, $i_k \neq i_m$ *iff* $k \neq m$, $i_m i_k = i_k i_m$, $i_k \in \mathbb{G}_n$

$$\mathbf{P}\left( \prod_{k=1}^{n} \cos(\theta_k) + \sum_{k=1}^{n} i_k \Big( \sin(\theta_k) \prod_{\ell=k+1}^{n} \cos(\theta_\ell) \prod_{m=1}^{k-1} e^{i_m \theta_m} \Big) \right) \tag{7}$$

$$\equiv \Big( \prod_{k=1}^{n} \cos(\theta_k) \Big) e_0 + \sum_{k=1}^{n} \Big( \sin(\theta_k) \prod_{\ell=k+1}^{n} \cos(\theta_\ell) \prod_{m=1}^{k-1} (-1)^{\frac{\theta_m - (\theta_m \bmod \pi)}{\pi}} \Big) e_k \tag{8}$$

Note the euclidean representation of $\mathbb{S}^{2^n-1}_{\mathbb{G}_n}$ is embedded in $\mathbb{R}^{2^n}$, which has a bijective mapping onto the Multicomplex numbers $\mathbb{C}_n$. This bijection can be shown by utilizing a forgetful functor to map the full Multicomplex basis set onto a set of orthonormal basis vectors for $\mathbb{R}^{2^n}$.

## 3 THE LGE FRAMEWORK

### 3.1 NORMALIZATION

The first step in the LGE dimensionality reduction algorithm is to normalize each point in the initial data set, and cache corresponding magnitudes separately from the generated unit vectors. This procedure serves the dual purpose of preparing the data for geometric dimensionality reduction from high-to-low dimensional spheres, as well as allowing uniqueness among vectors to be preserved by concluding the algorithm with a final rescaling by the original magnitude.

### 3.2 PRECONDITIONING

There are a number of measures that can be defined to tune the grouping fidelity of different types of data sets. The goal behind these metrics is to provide a range of measurements that can be used to quantitatively determine the optimal orientation of points in $\mathbb{R}^{2^n}$ for different clustering conditions.

#### 3.2.1 MINIMAL AVERAGE INNER PRODUCT (MAIP)

The most trivial metric introduced in this paper is the Minimal Average Inner Product, which measures the inner product between each normalized vector and the closest point on $\mathbb{S}_{\mathbb{G}_n}^{2^n-1}$ and averages the sum by the number of points. This allows for a computationally tractable method of optimizing the orientation of a dataset given only the original set and its corresponding set of contractions onto the surface of $\mathbb{S}_{\mathbb{G}_n}^{2^n-1}$.

**Definition 4 (Minimal Average Inner Product)**

$$\text{MAIP} = \frac{1}{m} \sum_{i=1}^{m} \langle \frac{x_i}{|x_i|}, \mathbf{P}(\frac{x_i}{|x_i|}) \rangle \tag{9}$$

$$x_i \in AX, \ A \subset SO(2^n), \ X \subset \mathbb{R}^{2^n}, \ \mathbf{P}(\frac{x_i}{\|x_i\|}) \in \mathbb{S}_{\mathbb{G}_n}^{2^n} \tag{10}$$

#### 3.2.2 BIJECTIVE WEIGHTED INNER PRODUCT (BWIP)

As we define more sophisticated tools, we include additional tuneable parameters that penalize vectors that neighbor non-reversible regions on $\mathbb{S}_{\mathbb{G}_n}^{2^n-1}$. Since all points in the Multicomplex rotation group with angle $\theta_k = \frac{\pi}{2}$ for $2 \leq k \leq n$ are non-reversible, we can define the measure: Bijective Weighted Inner Product, which specifically adds an angle proximity penalty to the overall sum.

**Definition 5 (Bijective Weighted Inner Product)**

$$\text{BWIP} = \frac{1}{m} \sum_{i=1}^{m} \langle \frac{x_i}{|x_i|}, \mathbf{P}(\frac{x_i}{|x_i|}) \rangle + \sum_{j=2}^{n} c_j |1 - \frac{\theta_j}{\frac{\pi}{2}}|^{r_j} \tag{11}$$

$$x_i \in AX, A \subset SO(2^n), X \subset \mathbb{R}^{2^n}, \ \mathbf{P}(\frac{x_i}{\|x_i\|}) \in \mathbb{S}_{\mathbb{G}_n}^{2^n}, \ r_j \in (0,1), \ c_j \in \mathbb{R}^+ \tag{12}$$

#### 3.2.3 TUNEABLE SINGULARITY AVOIDANCE MEASURE (TSAM)

The final measure comes from addressing the unique singularity with $\theta_1$ that arises due to the modularity shift in the LG fibration (Livschitz & Gu, 2022). By rewarding distribution clusters that avoid $\theta_1 = \pi$, we can improve clustering fidelity by preventing groups along the boundary from getting projected onto opposite orientations of the sphere.

**Definition 6 (Tuneable Singularity Avoidance Measure)**

$$\text{TSAM} = \frac{1}{m} \sum_{i=1}^{m} \langle \frac{x_i}{|x_i|}, \mathbf{P}(\frac{x_i}{|x_i|}) \rangle + c_1 |1 - \frac{\theta_1}{\pi}|^{r_1} + \sum_{j=2}^{n} c_j |1 - \frac{\theta_j}{\frac{\pi}{2}}|^{r_j} \tag{13}$$

$$x_i \in \mathbb{R}^{2^n}, \ \mathbf{P}\left(\frac{x_i}{|x_i|}\right) \in S_{\mathbb{G}_n}^{2^n-1}, \ r_j \in (0,1), \ c_j \in \mathbb{R}^+, \ 1 \leq j \leq n \tag{14}$$

In order to generalize LGE Dimensionality Reduction to higher dimensions, we must establish a contraction map from $S^{2^n-1}$ onto $S_{\mathbb{G}_n}^{2^n-1}$, as well as a projection map from $S_{\mathbb{G}_n}^{2^n-1}$ onto $S^n$. We can begin by defining a function that measures the distance between the manifold $S_{\mathbb{G}_n}^{2^n-1}$ and any given point in $\mathbb{R}^{2^n}$.

### 3.3 CONTRACTION MAP: SIDESTEPPING NEAREST NEIGHBOR SEARCH

One of the advantages to performing dimensionality reduction using geometric methods is the ability to reduce computationally costly processes with closed form analytical solutions. In order to find the exact closest point between any point in $\mathbb{R}^{2^n}$ and $S_{\mathbb{G}_n}^{2^n-1}$, we must setup a differential approach that minimizes inter-point distance. To define the distance map, we take the inner product of the difference between $\vec{S_{\mathbb{G}_n}^{2^n-1}} \subset S^{2^n-1}$ and any point in $\mathbb{R}^{2^n}$ with itself.

$$g(\theta_1, ..., \theta_n) = \langle \vec{S_{\mathbb{G}_n}^{2^n-1}} - \vec{w}, \vec{S_{\mathbb{G}_n}^{2^n-1}} - \vec{w} \rangle \tag{15}$$

$$= \langle \left(\prod_{k=1}^{n} \vec{e^{i_k\theta_k}}\right) - \vec{w}, \left(\prod_{k=1}^{n} \vec{e^{i_k\theta_k}}\right) - \vec{w} \rangle = 1 + \|w\|^2 - 2\langle \left(\prod_{k=1}^{n} \vec{e^{i_k\theta_k}}\right), \vec{w} \rangle \tag{16}$$

$$\vec{w} \in \mathbb{R}^{2^n} \tag{17}$$

In order to determine the closest point on $S_{\mathbb{G}_n}^{2^n-1}$ to $\vec{w}$, we must find all $\theta_k$ that minimize $g(\theta_1, ..., \theta_n)$. To do so, we differentiate $g$ with respect to each variable once independently, in order to setup the $n$ independent equations necessary for determining each $\theta_k$.

**Definition 7 (Sidestepping Nearest Neighbor in $\mathbb{R}^{2^n}$)** $\theta_1 \in [0, 2\pi), \ \theta_j \in [0, \pi), \ 2 \leq j \leq n$

$$\frac{\partial g}{\partial \theta_k} = -2\langle \frac{\partial \vec{S_{\mathbb{G}_n}^{2^n-1}}}{\partial \theta_k}, \vec{w} \rangle = -2\langle \left(i_k \prod_{k=1}^{n} \vec{e^{i_k\theta_k}}\right), \vec{w} \rangle = 0 \tag{18}$$

$$1 \leq k \leq n \tag{19}$$

The once computationally expensive discrete problem of determining the nearest neighbor of a point in a region has been transformed into a continuous optimization problem with fully defined constraints (Platt & Barr, 1987). This allows for either an immediate closed form solution such as for $n = 2$, or a computer-algebra based approach for determining $n$ unknowns in a system of $n$ non-linear equations.

### 3.3.1 CLOSED-FORM CONTRACTION MAP: FOUR DIMENSIONS

We can derive the closed form solution to the contraction map in four dimensions by taking advantage of the low dimensional trigonometric identities of $g(\theta_1, \theta_2)$ (Luna-Elizarrarás et al., 2015) given that $\vec{w} = (a, b, c, d)$.

$$\frac{1}{-2}\left(\frac{\partial g}{\partial \theta_1} + \frac{\partial g}{\partial \theta_2}\right) = (d-a)\sin(\theta_1 + \theta_2) + (b+c)\cos(\theta_1 + \theta_2) = 0 \rightarrow \tag{20}$$

$$\tan(\theta_1 + \theta_2) = \frac{b+c}{a-d} \tag{21}$$

$$\frac{1}{-2}\left(\frac{\partial g}{\partial \theta_1} - \frac{\partial g}{\partial \theta_2}\right) = -(a+d)\sin(\theta_1 - \theta_2) + (b-c)\cos(\theta_1 - \theta_2) = 0 \rightarrow \tag{22}$$

$$\tan(\theta_1 - \theta_2) = \frac{b-c}{a+d} \tag{23}$$

We can now take the $\tan^{-1}$ for each of these expressions, and complete our proof of the closed form solutions for the $\theta$ and $\theta_2$ that minimize the distance between $\mathbb{S}^3_{\mathbb{G}_2}$ and $S^3$.

$$\theta_1 = \frac{\tan^{-1}(\frac{c+b}{a-d}) + \tan^{-1}(\frac{b-c}{a+d})}{2} \tag{24}$$

$$\theta_2 = \frac{\tan^{-1}(\frac{c+b}{a-d}) - \tan^{-1}(\frac{b-c}{a+d})}{2} \tag{25}$$

We have now explicitly identified the $\theta$ and $\theta_2$ that minimize $g(\theta, \theta_2)$ in terms of the components of $w$. Formally stated, we have the following equality:

$$\min \| \langle (e^{i_1\theta_1} \cdot \vec{e^{i_2\theta_2}}) - (a, \vec{b, c}, d), \ (e^{i_1\theta_1} \cdot \vec{e^{i_2\theta_2}}) - (a, \vec{b, c}, d) \rangle \| =$$

$$\| (e^{\frac{i_1}{2}(\tan^{-1}(\frac{c+b}{a-d}) + \tan^{-1}(\frac{b-c}{a+d}))} \cdot \vec{e^{\frac{i_2}{2}(\tan^{-1}(\frac{c+b}{a-d}) - \tan^{-1}(\frac{b-c}{a+d}))}}) - (a, \vec{b, c}, d) \| \tag{26}$$

Accordingly, we define the closed-form contraction map from $S^3$ onto $\mathbb{S}^3_{\mathbb{G}_2}$. Note that the solution is magnitude-invariant, meaning that substituting $\frac{w}{\|w\|} \mapsto w$ does not impact the nearest neighbor result. This implies that the broader map from $\mathbb{R}^4$ onto $\mathbb{S}^3_{\mathbb{G}_2}$ can be defined as:

$$\mathbf{C}\big(\frac{(a, b, c, d)}{\|(a, b, c, d)\|}\big) = \big(e^{i_1\big(\frac{\tan^{-1}(\frac{c+b}{a-d}) + \tan^{-1}(\frac{b-c}{a+d})}{2}\big)} \cdot \vec{e^{i_2\big(\frac{\tan^{-1}(\frac{c+b}{a-d}) - \tan^{-1}(\frac{b-c}{a+d})}{2}\big)}}\big) \tag{27}$$

### 3.4 PROJECTION MAP: THE LG FIBRATION

The following definition was introduced in Livschitz & Gu (2022):

$$\mathbf{P}\big(\prod_{k=1}^n e^{i_k\theta_k}\big) \equiv \big(\prod_{k=1}^n \cos(\theta_k)\big)e_0 + \sum_{k=1}^n \big(\sin(\theta_k) \prod_{\ell=k+1}^n \cos(\theta_\ell) \prod_{m=1}^{k-1} (-1)^{\frac{(\theta_m - (\theta_m \, mod \, \pi))}{\pi}}\big)e_k \tag{28}$$

The projection maps each point in $\mathbb{S}^{2^{n}-1}_{\mathbb{G}_n}$ onto a polyspherical orthonormal representation of $S^n$ (Livschitz & Gu, 2022). This result can be used to generate the special case of $\mathbb{R}^4$ to $\mathbb{R}^3$ LGE dimensionality reduction.

#### 3.4.1 CLOSED-FORM PROJECTION MAP: FOUR TO THREE DIMENSIONS

We begin by setting $n = 2$ and using the LG fibration definition to generate the three dimensional output of the map:

$$\mathbf{P}\big(e^{i_1\theta_1} \cdot e^{i_2\theta_2}\big) = \cos(\theta_1)\cos(\theta_2)e_0 + \sin(\theta_1)\cos(\theta_2)e_1 + \sin(\theta_2)(-1)^{\frac{(\theta_1 - (\theta_1 \, mod \, \pi))}{\pi}}e_2 \tag{29}$$

This result provides us with the polyspherical definition of $S^2$ in terms of $\theta_1$ and $\theta_2$, the last definition necessary to explicitly define LGE over $\mathbb{R}^4$ (Koelink & Koornwinder, 1998). We will now proceed with reviewing the contraction and Projection map properties, as well as computing the special case of $\mathbf{P}(\mathbf{C}(S^3))$.

### 3.5 LGE: FOUR TO THREE DIMENSIONS

The first step of LGE dimensionality reduction is normalizing every data point in $\mathbb{R}^{2^n}$ and caching respective magnitudes for rescaling at the end of the procedure. Following normalization onto the

surface of the unit sphere $S^{2^n-1}$, each point is contracted onto $\mathbb{S}^{2^{n-1}}_{\mathbb{G}_n}$, projected onto $S^n$, then finally rescaled to $\mathbb{R}^{n+1}$. The entire LGE procedure for $\mathbb{R}^4$ can be computed as the following:

$$\mathbf{P}\big(\mathbf{C}\big(\frac{(a,b,c,d)}{\|(a,b,c,d)\|}\big)\big) = \cos(\theta_1)\cos(\theta_2)e_0 + \sin(\theta_1)\cos(\theta_2)e_1 + \sin(\theta_2)(-1)^{\frac{(\theta_1 - (\theta_1 \bmod \pi))}{\pi}}e_2 \quad (30)$$

$$\theta_1 = \frac{\tan^{-1}(\frac{c+b}{a-d}) + \tan^{-1}(\frac{b-c}{a+d})}{2}, \quad \theta_2 = \frac{\tan^{-1}(\frac{c+b}{a-d}) - \tan^{-1}(\frac{b-c}{a+d})}{2} \quad (31)$$

$$\mathbf{LGE}\big((a,b,c,d)\big) = \|(a,b,c,d)\| \cdot \mathbf{P}\big(\mathbf{C}\big(\frac{(a,b,c,d)}{\|(a,b,c,d)\|}\big)\big) \quad (32)$$

### 3.6 GENERALIZED LGE DIMENSIONALITY REDUCTION

We can summarize LGE dimensionality reduction algorithm as the following combination of maps:

$$\mathbf{C}(S^{2^n-1}) = \mathbb{S}^{2^n-1}_{\mathbb{G}_n}, \quad \mathbf{P}\big(\mathbb{S}^{2^n-1}_{\mathbb{G}_n}\big) = S^n \quad (33)$$

$$\mathbf{LGE}(\vec{x}) = \|\vec{x}\| \cdot \mathbf{P}\big(\mathbf{C}\big(\frac{\vec{x}}{\|\vec{x}\|}\big)\big) \quad (34)$$

$$\mathbb{R}^{2^n} \xrightarrow{\frac{\mathbf{x}}{\|\mathbf{x}\|}} S^{2^n-1} \xrightarrow{\mathbf{C}} \mathbb{S}^{2^n-1}_{\mathbb{G}_n} \xrightarrow{\mathbf{P}} S^n \xrightarrow{\|\mathbf{x}\| \cdot \frac{\mathbf{x}'}{\|\mathbf{x}'\|}} \mathbb{R}^{n+1} \quad (35)$$

## 4 EXPERIMENT

In this section, we highlight three examples that best compare t-SNE, UMAP, and LGE (van der Maaten & Hinton, 2008). Through each experiment we gain insight into the impact of dimensional compression on clustering distribution, as well as the effect of high-dimensional orientation on local and global spatial structure.

**Example 1:** A spherically distributed point cloud on the unit sphere in $\mathbb{R}^4$ centered at $(1,1,1,1)$. This single cluster projection illustrates qualitative preservation of local structure, roughly reproducing a spherical point cloud in $\mathbb{R}^3$.

**Example 2:** Two spherically distributed point clouds on unit spheres in $\mathbb{R}^4$ centered at $(1,1,1,1)$ and $(-1,-1,-1,-1)$. The dual cluster projection exercises orthogonal cluster separation *and* qualitative preservation of local structure.

**Example 3:** Five spherically distributed point clouds on unit spheres in $\mathbb{R}^4$ centered at $(1,1,1,1)$, $(1,0,0,1)$, $(2,2,0,1)$, $(-2,0,1,2)$, $(-1,-1,0,-1)$. This example illustrates the contortion of the $\mathbb{S}^3_{\mathbb{G}_2}$ manifold and the effect of optimizing orientation against a selected loss function.

*Hyperparameters and coefficients are detailed in Appendix A.*

### 4.1 QUALITATIVE EVALUATION

t-SNE and UMAP both rely on graph-based optimization techniques and typically initialize with a random embedding (McInnes et al., 2018). The LGE framework deterministically constructs embeddings that enable more faithful preservation of local structural relationships whereas the former tend to distort lower-density geometry (see Figure(s) 1 and 2).

In Figure 3, the t-SNE and UMAP projections exhibit compact clustering without overlap. Although the LGE projection delineates boundaries effectively, local substructure appears warped, reflecting distortions of the $\mathbb{S}^3_{\mathbb{G}_2}$ manifold. Since the LGE framework is *not* rotationally invariant, we have discussed various metrics for optimizing projection onto $\mathbb{S}^3_{\mathbb{G}_2}$ in Section 4.2.

Applying the MAIP loss function (via a gradient descent optimizer) reduces intra-cluster spread, leaving only limited overlap between adjacent clusters (see Figure 4). Further optimizing for contortions on $\mathbb{S}^3_{\mathbb{G}_2}$, we introduce BWIP and TSAM. While substructure preservation appears only moderately dense, cluster boundaries are more sharply defined. A perturbed variant of TSAM yields the

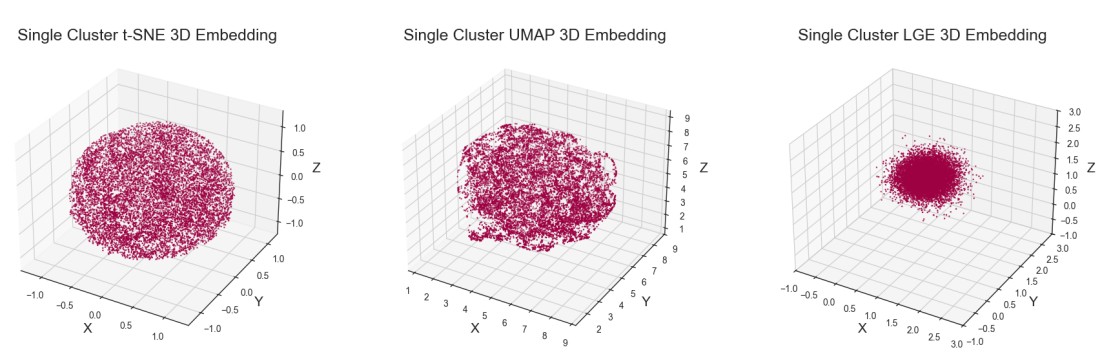

Figure 1: $\mathbb{R}^4 \mapsto \mathbb{R}^3$ single cluster projection of t-SNE, UMAP, and LGE respectively.

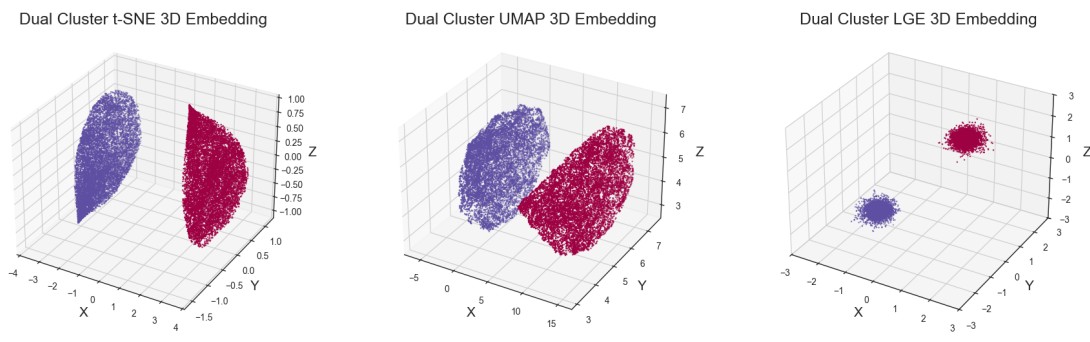

Figure 2: $\mathbb{R}^4 \mapsto \mathbb{R}^3$ dual cluster projection of t-SNE, UMAP, and LGE respectively.

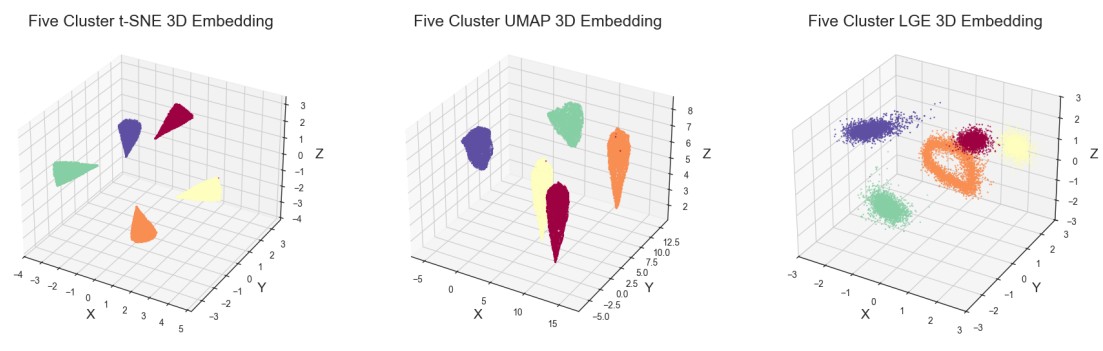

Figure 3: $\mathbb{R}^4 \mapsto \mathbb{R}^3$ five cluster projection of t-SNE, UMAP, and LGE respectively.

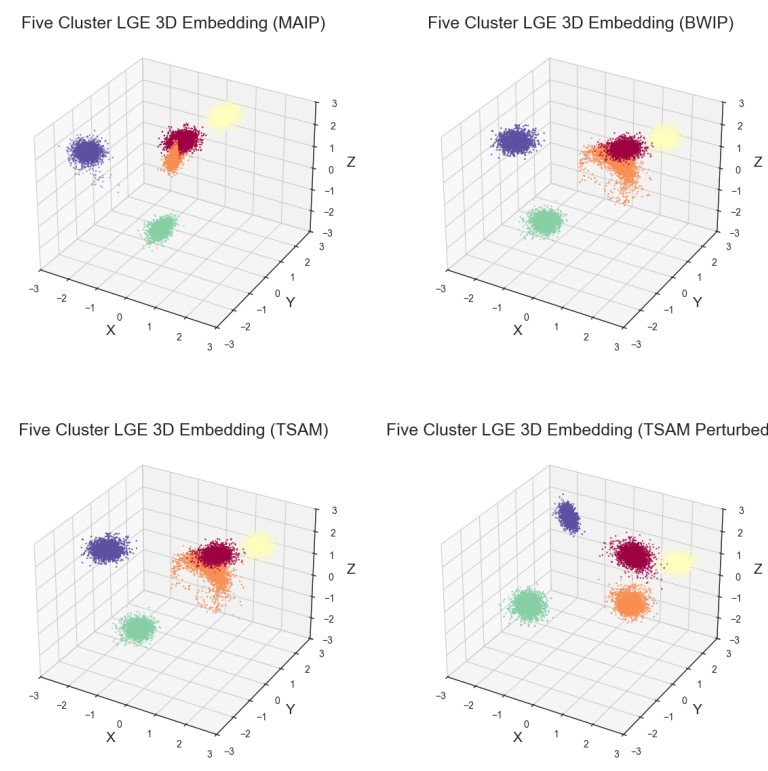

Figure 4: LGE orientation optimizer comparison.

most favorable result, exhibiting well-preserved geometry with clearly defined boundaries. The need for initial orientation perturbation likely arises from the non-trivial optimization landscape, analogous to user-defined hyperparameters such as perplexity in t-SNE or neighborhood size in UMAP (van der Maaten & Hinton, 2008; McInnes et al., 2018).

*Refer to Appendix B for details regarding the perturbed variant of TSAM.*

### 4.2 QUANTITATIVE EVALUATION

We now assess the quality of projection using the average inner product, silhouette score, and the Davies–Bouldin index (see Appendix C). We report the silhouette score, where values closer to 1 indicate well-separated and cohesive clusters (Rousseeuw, 1987), and the Davies–Bouldin Index, where lower values reflect more compact clusters with minimal overlap (Davies & Bouldin, 1979).

As shown in Table 1, the LGE framework outperforms both t-SNE and UMAP by more closely approximating the distribution of the unprojected, dual-cluster dataset. Though t-SNE and UMAP quantitatively outperform the *baseline* LGE framework when projecting the five-cluster dataset, the incorporation of preconditioning yields a consistent improvement in clustering quality (see Table 2).

### 4.3 TIME COMPLEXITY

As discussed in Section 3, the LGE framework iterates a maximum of $log(n)$ times until the target dimension, $n$, is reached. The time complexity is dominated by computation of the chosen loss function for $n$ samples and yields $\mathcal{O}(N)$ *per timestep*. As both $n$ and the maximum timestep remain constant, embedding generation complexity is simplified to $\mathcal{O}(N)$ (see Appendix D). The LGE framework enables efficient out-of-sample projection by re-using the cached rotation matrix followed by contraction and projection. This results in a $\mathcal{O}(1)$ upsert operation (see Appendix D).

Table 1: Quantitative Evaluation - Dual Cluster Statistics[†]

| Algorithm | Silhouette Score | Davies–Bouldin Index |
|---|---|---|
| Point cloud ($\mathbb{R}^4$) | 0.9085 | 0.1192 |
| t-SNE ($\mathbb{R}^4 \mapsto \mathbb{R}^3$) | 0.7559 | 0.3587 |
| UMAP ($\mathbb{R}^4 \mapsto \mathbb{R}^3$) | 0.7692 | 0.3332 |
| LGE ($\mathbb{R}^4 \mapsto \mathbb{R}^3$) | 0.9291 | 0.0923 |

[†]Two cluster dataset as described in Example 2.

Table 2: Quantitative Evaluation - Five Cluster Statistics[†]

| Algorithm | Silhouette Score | Davies–Bouldin Index |
|---|---|---|
| Point cloud ($\mathbb{R}^4$) | 0.8101 | 0.2492 |
| t-SNE ($\mathbb{R}^4 \mapsto \mathbb{R}^3$) | 0.8184 | 0.2657 |
| UMAP ($\mathbb{R}^4 \mapsto \mathbb{R}^3$) | 0.8009 | 0.2933 |
| LGE ($\mathbb{R}^4 \mapsto \mathbb{R}^3$) | 0.6858 | 0.6074 |
| LGE ($\mathbb{R}^4 \mapsto \mathbb{R}^3$) w/ MAIP Loss | 0.7761 | 0.2897 |
| LGE ($\mathbb{R}^4 \mapsto \mathbb{R}^3$) w/ BWIP Loss | 0.7355 | 0.4096 |
| LGE ($\mathbb{R}^4 \mapsto \mathbb{R}^3$) w/ TSAM Loss | 0.7344 | 0.4138 |
| LGE ($\mathbb{R}^4 \mapsto \mathbb{R}^3$) w/ TSAM Loss Perturbed | 0.8276 | 0.2280 |

[†]Five cluster dataset as described in Example 3.

## 5 DISCUSSION AND FUTURE WORK

The LGE framework provides a foundation for future dimensionality reduction techniques utilizing algebraic and topological symmetries. A key advantage is the $\mathcal{O}(1)$ time complexity for upsert operations. The framework may also provide a principled mechanism for enhancing the initialization phase of UMAP by introducing fixed *anchor points* within the low-dimensional embedding. Strategically anchoring selected data points could guide the UMAP optimizer toward configurations that better preserve local geometry (McInnes et al., 2018; Narayan et al., 2020).

A possible refinement involves examining the effects of affine transformations on the original dataset. In particular, preconditioning the dataset by moving all points into the positive orthant may yield higher-fidelity projections, as the $\mathbb{S}_{\mathbb{G}_n}^{2^n-1}$ manifold is likely better defined under these constraints. Furthermore, these constraints may help mitigate implicit contortions on the manifold and reduce the likelihood of projecting near or onto a singularity.

### 5.1 LIMITATIONS

In order to scale the functionality of LGE to high-dimensional applications, further work must be done to either determine a closed form solution for the Contraction map, or accelerate the "Nearest Neighbor" computation. We have shown a closed-form solution for the LGE map from $\mathbb{R}^4$ onto $\mathbb{R}^3$ in Section 3.5, however the closed-form projection for an arbitrary data point from $\mathbb{R}^{2^n}$ to $\mathbb{R}^{n+1}$ remains an open problem that relies on solving a system of $n$ non-linear equations. Deriving a closed-form solution would not only circumvent significant computational overhead, but would also improve the precision of data compression and reversibility.

There exist some promising computational methods that may effectively accelerate application of the contraction map, such as generating a dense cluster of surrounding points on $\mathbb{S}_{\mathbb{G}_n}^{2^n-1}$ and using $k$-nearest neighbor search to determine the nearest sampled neighbors (Bentley, 1975; Cover & Hart, 1967). Techniques such as DiskANN can also be employed to mitigate memory constraints, though practical computational bounds still remain (Jayaram Subramanya et al., 2019).

REPRODUCIBILITY STATEMENT

We are committed to ensuring reproducibility of our results. To this end, we will release the source code, hyperparameters, and exact instructions to reproduce all experiments if the paper is accepted. All datasets used will be made publicly available to facilitate independent verification. Further implementation and analysis details are provided in the appendix for completeness.

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

TECHNICAL APPENDICES AND SUPPLEMENTARY MATERIAL

## A  HYPERPARAMETER & COEFFICIENT DETAILS

Table 3: Experiment Hyperparameters

| Name | Value |
|---|---|
| Learning Rate | 0.1 |
| Maximum Iterations | 100 |
| Convergence Tolerance | 0.0001 |

The hyperparameters used with the gradient descent optimizer for all loss functions defined in Section 3.2 are listed in Table 3. These values were chosen in favor of computational runtime and convergence efficiency. Notably, the optimizer uses the "*Convergence Tolerance*" hyperparameter to stop iterating if the loss between iterations is less than the set value. This behavior prevents the optimizer from over-shooting local minima. Though for this paper simplicity was preferred, future experiments may find it beneficial to use an ADAM optimizer in lieu of a manual "*Convergence Tolerance*" (Kingma & Ba, 2014).

The loss function coefficients, $c_j$ and $r_j$, as shown in Section 3.2, were chosen to optimize for the quantitative metrics described in Section 4.2. MAIP (see Section 3.2.1) and BWIP (see Section 3.2.2) employed:

$$c_j = 0.1$$
$$r_j = 0.3 \tag{36}$$

Both TSAM (see Section 3.2.3) and the perturbed variant of TSAM utilized:

$$c_1 = 0.6$$
$$r_1 = 0.1$$
$$c_j = 0.1$$
$$r_j = 0.3 \tag{37}$$

## B  TSAM PERTURBED DETAILS

Table 4: TSAM Perturbed Initial Orientation

| $\theta_1$ | $\theta_2$ | $\theta_3$ | $\theta_4$ | $\theta_5$ | $\theta_6$ |
|---|---|---|---|---|---|
| 0.6983 | 1.3732 | 0.0466 | 1.2493 | 1.0139 | 0.3338 |

The perturbed version of TSAM follows the exact same methodology detailed in Section 3.2.3 but the initial orientation is randomly initialized instead of setting all $\theta_k$ to 0. In context of this paper and provided experiment results, the initial orientation was randomly initialized to the values shown in Table 4. The hyperparameters, such as the scaling coefficients, are kept consistent with those employed in the unperturbed variant.

## C  ADDITIONAL EXPERIMENT EVALUATION METRICS

Table 5: Five Cluster Average Inner Product

| Loss Function | Average Inner Product |
|---|---|
| Original Projection | 0.9069 |
| MAIP Loss | 0.9962 |
| BWIP Loss | 0.9647 |
| TSAM Loss | 0.9620 |
| TSAM Loss Perturbed | 0.9840 |

Table 6: Five Cluster Optimizer Terminal Orientation

| Optimizer | $\theta_1$ | $\theta_2$ | $\theta_3$ | $\theta_4$ | $\theta_5$ | $\theta_6$ |
|---|---|---|---|---|---|---|
| MAIP | 6.1585 | 0.0000 | 0.7200 | 0.5903 | 0.1366 | 6.2212 |
| BWIP | 0.3057 | 0.0013 | 0.0000 | 0.2276 | 0.0000 | 5.9819 |
| TSAM | 0.2616 | 0.0442 | 0.0000 | 0.2228 | 0.0000 | 6.0100 |
| TSAM Perturbed[†] | 6.2711 | 0.5916 | 0.0000 | 0.3356 | 0.0000 | 1.4041 |

[†]Initial conditions discussed in Appendix B.

Table 5 reports the mean inner product between each unit-normalized data point and its corresponding projection onto $\mathbb{S}^3_{\mathbb{G}_2}$. This metric serves as a loose indicator of *compression loss*, quantifying the deviation that arises when a data-point vector does not inherently reside on $\mathbb{S}^3_{\mathbb{G}_2}$. In particular, it captures the intra-vector angular discrepancy between the original vector and its nearest counterpart projected onto $\mathbb{S}^3_{\mathbb{G}_2}$. By applying various orientation optimizers as discussed in Section 3.2, a general upward trend in mean inner product is revealed, indicating a decreasing *compression loss*. Qualitatively and quantitatively this in-turn would result in an embedding generation that is more geometrically faithful to the original dataset.

Table 6 specifies the terminal orientation attained by the gradient descent optimizer with various loss functions for the five cluster dataset; expressed as a set of six angular parameters corresponding to a four-dimensional rotation.

## D  TIME COMPLEXITY COMPARISON

Table 7: Time Complexity Comparison

| Algorithm | Initial Embedding | Out-of-Sample Projection |
|---|---|---|
| t-SNE | $\mathcal{O}(N^2)$ | Not supported |
| UMAP | $\mathcal{O}(N \log N)$ | $\mathcal{O}(\log N)$ |
| LGE ($\mathbb{R}^4 \mapsto \mathbb{R}^3$) | $\mathcal{O}(N)$ | $\mathcal{O}(1)$ |
| LGE (generalized)[†] | $\mathcal{O}(N)^{†}$ | $\mathcal{O}(1)^{†}$ |

[†]Time complexity is *hypothetical*. (McInnes et al., 2018; Wei et al., 2020)

We have analyzed the time complexity of the LGE framework in Section 4.3 with a summary provided in Table 7 above. Furthermore, we present the hypothetical time complexity for the generalized LGE framework under the assumption that the limitations identified in Section 5.1 are effectively addressed.

For UMAP, the computational cost is dominated by generating the $k$-nearest-neighbor graph, yielding an effective complexity of $\mathcal{O}(N^{1.14}) \approx \mathcal{O}(N \log N)$ for practical values of $N$ (McInnes et al., 2018). Out-of-sample projection is bound by an approximate $k$-nearest-neighbor query; $\mathcal{O}(\log N)$ (Elkin & Kurlin, 2023).

In t-SNE, computing the full pairwise distance matrix dominates the computational cost, yielding an effective complexity of $\mathcal{O}(N^2)$ (Wei et al., 2020). Out-of-sample projection is not supported by t-SNE.

