# OpenReview forum: "Geometric Dimensionality Reduction: Framework and Open Challenges"
_ICLR.cc/2026/Conference — ICLR 2026 Conference Withdrawn Submission_

### Official Review · Reviewer_KcrM · 2025-10-29

**Soundness:** 2
**Presentation:** 3
**Contribution:** 2
**Rating:** 4
**Confidence:** 3

**Summary:**

This paper introduces LGE (Logarithmic Geometric Embedding), a deterministic dimensionality reduction framework that maps data from $\mathbb{R}^{2n}$ to $\mathbb{R}^{n+1}$ through two analytic mappings: a contraction $C$ that projects each normalized point onto a structured “multicomplex unit manifold” $S^{2n-1}_{G_n}$ and a projection $P$ that transforms that manifold point into a lower-dimensional representation. The authors derive closed-form expressions for $n=2$ and propose three rotation-based objectives (MAIP, BWIP, TSAM) for global orientation optimization. Synthetic experiments show that LGE can separate clusters comparably or better than t-SNE and UMAP on simple examples while being fully analytic and allowing direct out-of-sample mappings.

**Strengths:**

- The method is novel conceptually, combining multicomplex algebra with dimensionality reduction in an analytic, deterministic pipeline.
- The closed-form derivation for the $\mathbb{R}^4 \to \mathbb{R}^3$ case is mathmatically interesting and produces interpretable geometric mappings.
- The paper emphasizes out-of-sample embedding without training, which is appealing compared to t-SNE.
- The writing is detailed, and the orientation objectives show creative attempts to stabilize the mapping and improve separability.

**Weaknesses:**

- For $n>2$, the contraction requires solving $n$ nonlinear equations with no algorithmic details or computational analysis. The complexity and numerical stability of that step remain open.
- The paper assumes that vectorization from multicomplex coordinates to $\mathbb{R}^{2n}$ preserves the Euclidean norm, but does not prove it. A short derivation showing that $\|\text{vec}(\prod e^{i_k \theta_k}) \|=1$ under the chosen inner product would make the contraction objective formally sound.
- Experiments are limited to synthetic 4D clusters with spherical Gaussian-like clouds. This makes the geometry particularly friendly for a spherical manifold contraction. Currently, Table 1 shows LGE outperforming both methods on a two-cluster toy case; Table 2 shows LGE underperforming on five clusters unless orientation is carefully tuned, which weakens the general claim that LGE preserves local structure “more faithfully.”
- To support claims relative to t-SNE/UMAP, include real datasets and neighborhood metrics used in the dimension reduction literature (e..g, KNN score) in addition to silhouette and Davies–Bouldin.
- Also report run time, parameter configurations for the baselines (e.g., n_neighbors, min_dist), and variance over runs.

**Questions:**

See weaknesses. I would be willing to raise my score if the authors address the concerns above.

---

### Official Review · Reviewer_Yxun · 2025-10-29

**Soundness:** 1
**Presentation:** 1
**Contribution:** 1
**Rating:** 2
**Confidence:** 3

**Summary:**

The paper introduces a dimensionality reduction framework wherein closed-form projection from $\mathbb{R}^4$ to $\mathbb{R}^3$ is possible. Furthermore, the authors show that through that framework dimensionality reduction from $\mathbb{R}^{2^n}$ to $\mathbb{R}^{n+1}$ is possible by solving a system of $n$ non-linear systems. Experimentally, the authors evaluate the projection from 4d to 3d on toy data sets, comparing to established methods (UMAP and t-SNE).

**Strengths:**

The paper introduces a methodology for dimensionality reduction that takes an interesting new perspective, using a clever reparameterisation into the simple multicomplex rotation group.

**Weaknesses:**

The paper is very bluntly written. Specifically, each section introduces a series of definitions and moves on to the next point, with no preparation or overall underlying "thread" connecting all of them. I am unsure if I understood the methodology in its entirety. An Algorithm listing or similar would be very helpful in that regard. I understand that Eq. (2), (34), and (35) summarise it in steps, but the details of each step are obscured in the many definitions and generalisations which are not employed.

As another example, the notation and language is extremely confusing. The symbol $\mathbb{S}_{\mathbb{G}_n}^{2^{n-1}}$ seems to denote a group, but is used as a point in Eq. (15). It is unclear how $g()$ is supposed to be understood as: the phrasing talks about sidestepping nearest neighbour search, but seems to be specifically about individually mapping points.

A substantial weakness of the method is the lack of general applicability in its natural form. The closed-form projection from $\mathbb{R}^4$ to $\mathbb{R}^3$ is intellectually nice, but limited in applicability. Meanwhile, the first sentence of the abstract and paragraph of the introduction would suggest a generally-applicable method.

I am aware that even practical use of t-SNE frequently needs an initial step with PCA to reduce the dimensionality of the input to a few dozen, then initialising the algorithm with the resulting mapping (of course whenever good distances are; see its implementation in the popular scikit-learn Python package, for instance). Still, I do not see the value of the contribution for any practical application, since reduction from $\mathbb{R}^4$ to $\mathbb{R}^3$ is hardly a challenging problem.

Overall, in its current form, I believe the paper is not ready for publication.

**Questions:**

1. Why are Eqs. (2) and (3) restated as Eqs. (34) and (35)?
2. Since automatic differentiation is so prevalent nowadays, why do the authors dedicate space in the main paper to stating $\frac{\partial g}{\partial \theta_k}$? Is there a specific benefit from knowing its analytical form?
3. Closely related to the previous section: the steps in deriving the closed-form solution don't really point useful insights from the process. Was there a point in making them explicit?
4. The paper does not discuss the meaning of norms from vectors in $\mathbb{R}^{2^n}$ being used to rescale vectors mapped down to $\mathbb{R}^{n+1}$. Can these be considered meaningful rescaling operations? I assume that, as $n$ increases, this will become more and more problematic to interpret.

Other comments:
- No display-style (on their own line) equations are part of the running text
- Definitions are too simple to require their own environments/paragraphs. Perhaps switch over to running text as it suffices.
- l. 18: axis => axes
- l. 85: Sentence starts with math symbol
- l. 106: euclidean => Euclidean

---

### Official Review · Reviewer_aiHn · 2025-10-31

**Soundness:** 2
**Presentation:** 2
**Contribution:** 2
**Rating:** 2
**Confidence:** 3

**Summary:**

This paper proposes Geometric Dimensionality Reduction, a framework that performs logarithmic data compression for dimensionality reduction. The framework consists of two main components: contraction and projection. The contraction map transforms a vector into a manifold, while the projection map further maps this manifold onto a lower-dimensional surface. The paper focuses on the case where the manifold is defined by the Simple Multicomplex Rotation Group, introducing several mathematical concepts necessary to describe this structure. Various formulations of the contraction map are studied, with a special emphasis on the case of $\mathbb{R}^4\mapsto \mathbb{R}^3$. Experiments on this case include qualitative evaluations and computational time analyses. Limitations are discussed.

**Strengths:**

The paper introduces a mathematically original approach to dimensionality reduction that differs from existing graph-based or probabilistic methods like t-SNE or UMAP.

It uses multicomplex algebra and manifold geometry to construct explicit contraction and projection maps. I think this is conceptually unique and rarely explored in current literature.

The proposed idea bridges algebraic geometry and machine learning, which could inspire future interdisciplinary research.

**Weaknesses:**

The proposed method is limited in scope, as it only supports dimensions that are $\mathbb{R}^{2^n}\mapsto \mathbb{R}^{n+1}$, restricting its practical applicability.

The motivation for why this algebraic structure leads to a better dimension reduction is not fully demonstrated.

The empirical evaluation is narrow and does not convincingly demonstrate the practical advantage of the proposed approach over existing methods.

**Questions:**

1. One major concern lies in the dimensionality constraint of the proposed method. The framework appears limited to mappings of the form $\mathbb{R}^{2^n}\mapsto \mathbb{R}^{n+1}$, which restricts its applicability to practical settings where dimensionality reduction is typically performed from arbitrary $\mathbb{R}^{n}\mapsto \mathbb{R}^{k}$.
   Could the authors elaborate on why this “logarithmic data compression” formulation is an important or necessary problem to consider? In particular, what motivates focusing on this specific power-of-two structure, and how might the proposed framework be generalized to handle arbitrary input and output dimensions?

2. I am not fully convinced about the practical benefits of the proposed framework. Possible advantages may include the ability to reconstruct the original data from its compressed representation and the deterministic nature of the out-of-sample projection. However, these seem relatively mild. Demonstrating a real application that leverages these advantages would substantially strengthen the paper.
In this regard, I also have several specific questions:

a) In the abstract, it is stated that the reduced axes in t-SNE and UMAP are non-interpretable. Are the axes in the proposed method interpretable?

b) The paper mentions that hyperparameter tuning in t-SNE and UMAP is non-trivial, yet Section 3 introduces tunable parameters in MAIP and BWIP. Could you elaborate on how these parameters are tuned, and whether such tuning leads to empirical improvements?

c) The empirical results indicate that TSAM Loss Perturbed performs best. Could you clarify what this procedure entails and explain why it outperforms MAIP, BWIP, and the baseline TSAM Loss?

3.   For potential readers’ clarity, I believe the algorithmic details of the projection step should be presented more explicitly. Equation (28) states that the projection maps each point in $\mathbb{S}^{2^{n-1}}_{\mathbb{G}_n}$

 onto a polyspherical orthonormal representation of $S^n$ (Livschitz \& Gu, 2022), but the definition of $mod \pi$ is not specified. Does this imply that, given the contracted representation $\mathbb{S}^{2^{n-1}}_{\mathbb{G}_n}$the projection can be computed efficiently in the general case (with arbitrary $n$)? If so, could you clarify the computational complexity of this step? Also, $N$ appears in the time-complexity discussion, but its definition is not explicitly provided. Could you clarify what  $N$ denotes in this context?

---

### Official Review · Reviewer_RE9j · 2025-10-31

**Soundness:** 2
**Presentation:** 2
**Contribution:** 1
**Rating:** 2
**Confidence:** 4

**Summary:**

The paper describes a framework for implementing geometric dimensionality reduction named LGE after the LG fibration. It starts by describing the general process for applying LGE which consists of the composition $\mathbb{R}^{2^n}\rightarrow S^{2^n - 1}\rightarrow \mathbb{S}^{2^n-1}_{\mathbb{G}_n}\rightarrow S^n\rightarrow \mathbb{R}^{n + 1}$ of mappings. The first mapping is a scaling of vectors by their norms to project them to a hypersphere, the second one, $C$ a contraction, the third, $P$, the fibration projection and the last one a scaling by the ratio of the norms of the vector in the original space to the projection space.

They go on to define different optimization objectives for the space of $SO(2^n)$ transforms of the dataset. The objectives are used to minimize the inner products of the normalized vectors to their contraction with optional additional terms which penalize solutions close to points where the fibration projection is not invertible or singular. The fibration projection has a fixed closed form and the contraction is would ideally be the mapping of points in $\mathbb{R}^{2^n}$ to the closest point to them in $\mathbb{S}^{2^n-1}_{\mathbb{G}_n}$. For the case of $\mathbb{R}^4$, a closed formula is derived for $C$.

Three synthetic datasets comprised of spherically distributed point clouds in $\mathbb{R}^4$ with one, two and five clusters are considered. There LGE projection are visually compared and scored against t-SNE and UMAP on the Silhouette score and the Davies-Bouldin index.

**Strengths:**

- The paper helps a bit to better understand the "The LG Fibration" by Livschitz & Gu which introduces the idea of projections based on the LG fibration.

**Weaknesses:**

- The novelty of the paper is relatively limited. The largest is material from "The LG Fibration" by Livschitz & Gu and the paper implements the method described there. Even the closed formula at line 240 which minimizes the distance of a point to $\mathbb{S}^{3}_{\mathbb{G}_2}$ is the same as the one in the original paper when replacing $(a, b, c, d)$ with their polyspherical coordinates and does the computations (though the original paper does not explicitly mention the distance minimization).

- The aforementioned paper it is based on has not been published in any peer reviewed conference or journal, making the full review of the current paper hard as one practically needs to verify that both are scientifically sound.

- The experiments are extremely limited. The spherical point clouds are very well suited for such a projection and do not really provide evidence for the method. Though one is limited to datasets in $\mathbb{R}^4$, there are multiple mathematical examples of manifolds like tori, projective spaces etc. and mixtures of them, but one could also more realistic datasets often used as benchmarks for dimensionality reduction methods such as COIL20, MNIST, FMNIST, Google news etc. by just projecting them first to $\mathbb{R}^4$ with PCA. This way one could also test more appropriate metrics like trustworthiness or nearest neighbor accuracy.

- I think the method should also be compared to PCA. In some sense PCA is the linear counterpart of the presented algorithm as it also looks for an $SO(2^n)$ transform of the dataset which minimizes its distance from a hyperplane instead of a manifold. This would server as good benchmark as it has more of the flavor of a rigid projection and it extrapolates better than non-linear methods (though of course it is not that flexible).

- Again regarding benchmarking. In the appendix you mentioned that the hyperparameters of BWIP and TSAM are optimized for the metrics you are using for scoring. This on the one hand makes the comparison much less reliable and on the other hand raises the risk of having parameters which need to be tuned for any new dataset. If you think this is not the case, you should probably add an ablation testing the sensitivity of changes to their values.

**Questions:**

- line 135, 150, 161: In the formulas of the optimization objectives do you mean $C(\frac{x_i}{|x_i|})$ instead of $P(\frac{x_i}{|x_i|})$? The latter does not make sense and it value is not in $\mathbb{S}^{2^n}_{\mathbb{G}_n}$.

- In section 3.2 you never make it clear what you are optimizing for. From the context and what I've seen in similar works, I assume that you are looking for a rotation matrix $A$ which minimizes the objective.

- line 193: Here, among other places, you are using $\mathbb{S}^{2^n}_{\mathbb{G}_n}$ as a variable representing some element of the manifold. This notations is very confusing, I think it would be preferable to just use a separate variable name.

---

### Note · Authors · 2025-11-30

**Comment:**

Dear Reviewers,

We wanted to thank you for your diligent review of our prospective mathematical & academic submission.

As we seek to improve our paper and meaningfully contribute to the cutting edge applications of topological data analysis, we will in particular work to convert the general case of our logarithmic compression methodology into an algorithmic structure. This repeated note made it clear that this structure will improve both with the clarity of the research contribution, as well as application readiness by researchers & industry alike.

Thank you once again for providing our team with the necessary feedback to grow as researchers, and I wish you a wonderful upcoming holiday season.

**Withdrawal Confirmation:**

I have read and agree with the venue's withdrawal policy on behalf of myself and my co-authors.